# Self supervised learning for in vivo localization of microelectrode arrays using raw local field potential

Tianxiao He[1,*]    Malhar Patel[1,*]    Chenyi Li[1]    Anna Maslarova[2]    Mihály Vöröslakos[2]
Nalini Ramanathan[1]    Wei-Lun Hung[1]    György Buzsáki[2]    Erdem Varol[1,2]

[1]Department of Computer Science, New York University
[2]Neuroscience Institute, Grossman School of Medicine, New York University

## Abstract

Recent advances in large-scale neural recordings have enabled accurate decoding of behavior and cognitive states, yet decoding anatomical regions remains underexplored, despite being crucial for consistent targeting in multiday recordings and effective deep brain stimulation. Current approaches typically rely on external anatomical information, from atlas-based planning to post hoc histology, which are limited in precision, longitudinal applicability, and real-time feedback. In this work, we develop a self-supervised learning framework, **Lfp2vec**, to infer anatomical regions directly from the neural signal in vivo. We adapt an audio-pretrained transformer model by continuing self-supervised training on a large corpus of unlabeled local-field-potential (LFP) data, then fine-tuning for anatomical region decoding. Ablations show that combining out-of-domain initialization with in-domain self-supervision outperforms training from scratch. We demonstrate that our method achieves strong zero-shot generalization across different labs and probe geometries, and outperforms state-of-the-art self-supervised models on electrophysiology data. The learned embeddings form anatomically coherent clusters and transfer effectively to downstream tasks like disease classification with minimal fine-tuning. Altogether, our approach enables zero-shot prediction of brain regions in novel subjects, demonstrates that LFP signals encode rich anatomical information, and establishes self-supervised learning on raw LFP as a foundation to learn representations that can be tuned for diverse neural decoding tasks. Code to reproduce our results is found in the github repository at `https://github.com/tianxiao18/Lfp2vec`.

## 1 Introduction

The ability to precisely localize high-density multielectrode recording sites *in-vivo* is a critical first step for systems and cognitive neuroscience research. Without accurate knowledge of which brain layer or sub-region each channel captures, subsequent analysis of cell-type specificity [1], circuit motifs [2], or disease biomarkers [3] become unreliable. In clinical settings such as deep brain stimulation at focal sites, the precision of electrode placement can significantly impact therapeutic outcomes and cognitive changes post-operation [4]. Today, most laboratories plan probe trajectories pre-operatively using brain atlas coordinates [5] or MRI scans [6], and validate them post-operatively by follow-up scans or *ex-vivo* histology [7]. However, these approaches lack in vivo knowledge of probe locations, and they are labor intensive, error-prone (limited by imaging resolution) and unsuitable for chronic recordings, flexible probes or clinical settings where tissue cannot be sacrificed.

---

[*]denotes equal contributions.

39th Conference on Neural Information Processing Systems (NeurIPS 2025).

As multishank arrays scale to over a thousand channels [8], both the data collection throughput and the need for accurate localization performance increases.

Recent work has attempted to address this using spike-based localization with action potential waveforms [9] or firing rates [10, 11]. However, these approaches require potentially inefficient and error-prone spike sorting [12] and ignore the rich structure in the local field potential (LFP) band [13, 14]. In parallel, self-supervised learning has been applied to unlabeled time series and various downstream tasks in audio [15] and neural signals [9, 16, 17]. Most existing work decodes behavioral states [18], discrete events [19], or speech [20] from LFP, rather than anatomical mapping. General purpose time series SSL models are typically designed with temporal prediction objectives [21], but they ignore anatomical priors and laminar continuity that characterize linear probes, which limits their use for brain region localization [22]. To our knowledge, no existing study has applied self-supervised learning to raw LFP signals, nor tested whether such models can generalize across species, probe types, and labs for scalable and reproducible brain region localization.

To this end, we introduce **Lfp2vec**, a self-supervised framework for anatomical localization from raw LFP recordings. Our approach adapts from wav2vec 2.0 [21], a foundation model pre-trained on audio signals, to the domain of electrophysiology. Using contrastive learning with masked prediction, we train on raw voltage traces without labels or probe geometry. A key aspect of our approach is transfer learning across modalities: instead of training from scratch, we initialize the model using wav2vec 2.0 and continue self-supervised training on large-scale LFP data. We hypothesize that representations learned from general purpose time series data are partially transferable to the neural domain. As our ablation studies will show (Fig. 5), this cross-domain initialization provides a substantial performance boost over random initialization, highlighting the value of out-of-domain foundation models for biomedical applications.

We then evaluate the output embeddings on recordings from four labs: Allen Institute [23], International Brain Lab [24], Neuronexus SiNAPs [8], and Neuropixel-NHP with macaque reaching dataset. We find that the embeddings cluster by anatomical subregion across sessions and animals, and can be decoded with high accuracy using multilayer perceptron. Beyond localization, these embeddings are also transferable to downstream tasks such as Alzheimer's disease classification from hippocampal dynamics. Together, our results show that neural population activity encodes brain region identity, and highlight the potential of self-supervised model to handle noisy and non-stationary neural time series data.

The contributions of this work include:

- A novel application of self-supervised learning to LFP signals, extracting embeddings for multiple downstream tasks.

- Empirical evidence that raw LFP signals encode rich anatomical and functional information, with SSL extracted features outperforming handcrafted features.

- A robust automatic probe localizer that achieves high-accuracy zero-shot prediction on held-out subjects and sessions, and generalizes across probe geometries, labs, and species (rodent to non-human primate).

## 2 Related Work

### 2.1 Self Supervised Learning in Extracellular Recordings

Self-supervised learning (SSL) has become a powerful tool in both machine learning and neural data science, enabling generalization across a wide range of downstream tasks. Models such as wav2vec 2.0 [21], timesFM [15] learn latent structure from time series via masked prediction, but they are not tailored to extracellular recordings and overlook domain-specific challenges such as biological noise, non-stationarity, and diverse sampling rates. In neural data science, SSL has shown promise primarily in the spike domain. Methods such as POYO [9], NEMO [11], NDT2 [17], and NEDS [16] use contrastive or masked objectives to learn embeddings from spiking activity. However, these methods typically rely on firing rates, spike times, or waveform features that are sensitive to spike sorting quality [25], and do not extend to local field potentials which are more broadly accessible. Models like BrainBERT [20], Brant [26] and related intracranial models [27] use broadband signals to decode

speech and cognitive states, but are not designed for anatomical localization. While effective, they do not incorporate anatomical priors or evaluate spatial generalization.

## 2.2 Brain Region Decoding

Traditional brain region decoding often relies on hand-crafted electrophysiological features or biomarkers. For example, sharp wave ripples help identify CA1 sublayers in the hippocampus [28], gamma coherence distinguishes hippocampal subregions [29], and firing statistics (e.g., burstiness, firing rates) differentiate cortical layers in V1 [30]. However, these approaches are typically heuristic, region-specific, and not learned from data. More recent methods apply deep learning on firing rates and spike timing to infer anatomical identity [31], but still rely on manual feature design and dense region labels, limiting scalability. Self-supervised models like NEMO [11] learn neuron embeddings for region decoding at single-unit level, yet they ignore population-level dynamics that captures brain region identity. Furthermore, most models are trained and evaluated within a single lab and do not evaluate generalization across labs and species, despite known variability in surgical procedures, probe designs, and behavioral paradigms [32, 24]. To address these limitations, we apply self-supervised learning to raw, multi-channel LFP recordings to learn anatomically meaningful representations that generalize across labs and recording setups.

## 3 Datasets

For brain region decoding, we apply two publicly available datasets and two private datasets across different species, recording probes, and experimental setup.

**IBL Reproducible Electrophysiology Dataset.** This dataset consists of Neuropixels recordings from mice performing visual decision-making task [24], collected by International Brain Lab (IBL). Ground-truth anatomical labels were obtained through post hoc histology by registering fluorescent probe tracks to the Allen Common Coordinate Framework. Labeled brain regions include secondary visual cortex (VISa), hippocampal subregions (CA1, CA3, dentate gyrus), and thalamic nuclei (LP, PO).

**Allen Visual Coding Dataset.** This dataset consists of Neuropixels recordings from mice performing a visual stimulus presentation task [23]. Recordings were sampled at 1250 Hz and span regions across the visual cortex, hippocampus (CA1, CA2, CA3, dentate gyrus), and thalamus. Ground-truth labels were obtained via post hoc histological analysis.

**Neuronexus Multi-Shank Dataset.** This dataset consists of 1024-channel Multi-shank Simultaneous Neural Active Pixel Sensor (SiNAPS) probe recordings from mice during spontaneous activity [8]. It includes five transgenic mice modeling Alzheimer's Disease (APP/PSEN1) and two normal controls. Recordings target hippocampal subregions including CA1, CA2, CA3, dentate gyrus, and surrounding cortex. The SiNAPs probe has 8 shanks with 128 channels per shank, enabling denser spatial coverage of the hippocampus than standard Neuropixels probes. Anatomical labels were assigned by expert annotation using electrophysiological landmarks and manual registration to a brain atlas.

**Macaque Sequential Reaching Dataset.** This dataset consists of Neuropixels-NHP recordings from a macaque monkey performing center-out and sequential reaching tasks. Recordings target motor-related regions including the basal ganglia, supplementary motor area (SMA), and primary motor cortex (M1). Ground-truth region labels were obtained by registering a 3D-printed electrode grid to MRI data and the recording chamber.

## 4 Methods

### 4.1 Preprocessing

To reduce non-physiological artifacts, we preprocess raw LFP recordings using the International Brain Lab (IBL) destriping pipeline [33]. Signals are zero-phase high-pass filtered at 2 Hz to remove

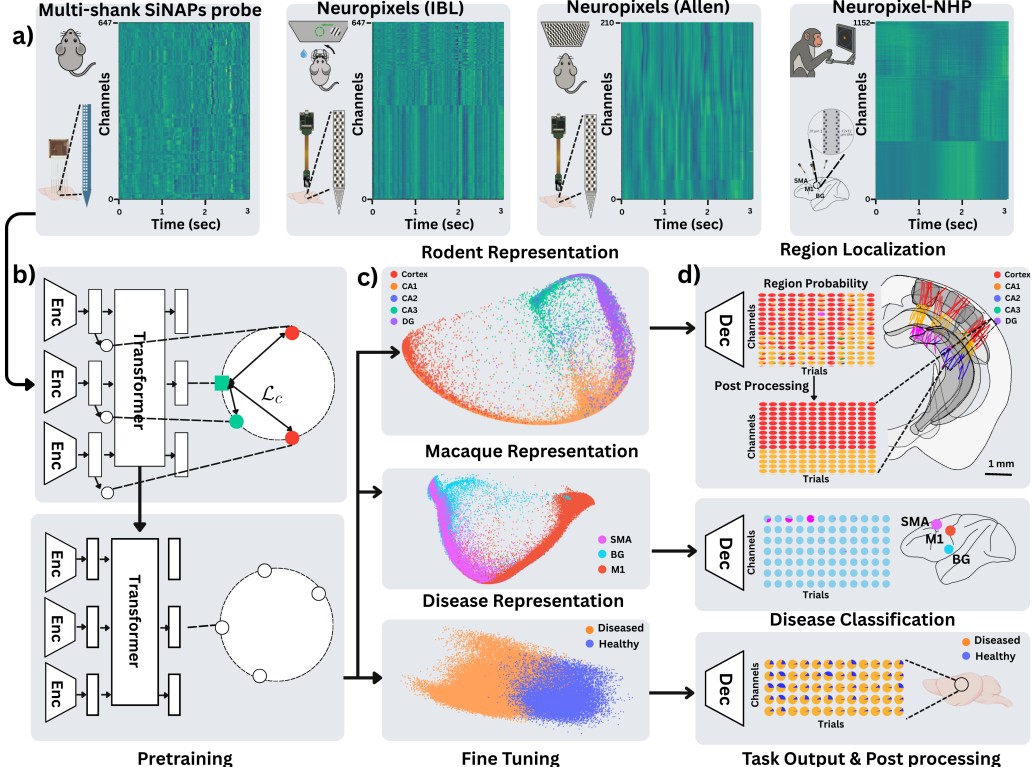

Figure 1: **Overview of the Lfp2vec pipeline**. a) Example multi-species local-field-potential (LFP) segments: SiNAPs (mouse), Neuropixels–IBL (mouse), Neuropixels–Allen (mouse), and Neuropixels-NHP (Macaque), segmented into short spatiotemporal windows. b) Each window is encoded using a 1-D convolutional encoder and a Transformer, trained with a masked-prediction contrastive objective to learn spatially coherent representations. The pretrained model is then fine-tuned for downstream tasks. c) Learned embeddings after fine-tuning: rodent hippocampus (top), macaque motor regions (middle), and healthy vs. disease-model recordings (bottom). Colors denote ground-truth anatomical or experimental labels. d) Embeddings drive lightweight decoders for downstream tasks: brain-region localization (top: rodent, middle: macaque) and session-level disease classification (bottom; Alzheimer's model vs. healthy). Each dot shows predicted region probabilities pie charts per trial. Columns correspond to trials, rows to channels, and colors represent brain regions or disease labels in c).

slow drifts, bad channels are detected using amplitude and spectral criteria and interpolated, and common-mode noise is removed by subtracting the per-sample median across valid channels. The cleaned recordings are segmented into 3-second trials and normalized (Fig. 1a), which standardizes inputs across devices (e.g., Neuropixels vs. Neuronexus) and species (mouse vs. macaque). In practice, model performance is sensitive to preprocessing and signal quality; insufficient artifact removal or quality control can substantially degrade representation quality and downstream accuracy.

## 4.2 Model Architecture

We adapt the wav2vec 2.0 framework [21] to LFP recordings, as shown in Fig. 1b. Rather than handcrafted spectral features (e.g., band power or spectrograms), we learn tokens directly from raw LFP by segmenting the continuous signal and encoding each window with a 7-layer 1D CNN for each channel and trial. The resulting tokens are masked and passed to a 12-layer Transformer context network, which predicts the correct target among distractors at masked time steps using surrounding context. For downstream decoding, we mean-pool Transformer outputs across time and apply a two-layer MLP classifier to predict brain-region or disease labels (Fig. 1d).

## 4.3 Objectives

We train **Lfp2vec** in two stages, adapting from a pretrained self supervised model to LFP recordings, as shown in Figure 1 b. The objective closely follows the wav2vec 2.0 contrastive learning framework.

**Self-supervised pretraining.** Given an LFP segment $x \in \mathbb{R}^T$, we compute tokens $z = E(x)$ and discretize them via Gumbel-softmax quantization $q = Q(z)$ with multiple codebooks [21]. We randomly mask time steps $\mathcal{M}$ in $z$, feed the masked latents $\tilde{z}$ into the Transformer, and obtain contextual embeddings $c = C(\tilde{z})$. For each masked step $m \in \mathcal{M}$, we apply a contrastive objective that pulls the context token $c_m$ (Transformer output at $m$) toward the target token $q_m$ (quantized vector at $m$), while pushing it away from distractors $\{q_k\}_{k \neq m}$ sampled from other time steps in the same batch. With temperature $\tau$, the loss is:

$$\mathcal{L}_{\text{SSL}} = -\sum_{m \in \mathcal{M}} \log \frac{\exp\big(\text{sim}(c_m, q_m)/\tau\big)}{\sum_{k \in \mathcal{K}(m)} \exp\big(\text{sim}(c_m, q_k)/\tau\big)}.$$

where $\mathcal{K}(m)$ denotes the set of distractors unioned with the true quantized vector $q_m$. To encourage diverse codebook usage, we additionally apply a diversity loss $\mathcal{L}_{\text{div}}$ weighted by $\lambda$. The overall pretraining objective is $\mathcal{L}_{\text{SSL}} + \lambda\mathcal{L}_{\text{div}}$ [21].

**Supervised fine-tuning.** We attach a classification head $h(\cdot)$ to the pooled context outputs $\bar{c} = \text{POOL}(c)$, and fine-tune all parameters end-to-end on anatomical region labels $\{y_i\}$. Denoting model outputs as $\hat{y}_i = h(\bar{c}_i)$, we minimize cross-entropy: $\mathcal{L}_{\text{CE}} = -\sum_i y_i^\top \log \hat{y}_i$, allowing the network to refine its representations for accurate region decoding.

## 4.4 Post processing

To improve decoding performance, we incorporate temporal and spatial priors via a lightweight post-processing pipeline (Figure 1 d). We first smooth model's per-timepoint predictions by averaging class probabilities over a temporal window $\mathcal{W}_t$ with class prior $\pi_c$, and assign the final label as $\hat{y}_t = \arg\max_c \frac{1}{|\mathcal{W}_t|} \sum_{t' \in \mathcal{W}_t} p_{t',c} \pi_c$., which suppresses transient fluctuations and abrupt label changes. We then enforce spatial continuity by assigning each channel the majority label of its 5 nearest neighbors, smoothing out isolated misclassifications. This step leverages the strong anatomical prior that adjacent channels on a linear probe are highly likely to reside in the same or contiguous brain regions. As shown in Figure 2d, these two steps improve channel-wise location prediction without retraining or requiring anatomical atlases, and run in milliseconds per session.

# 5 Experiments

## 5.1 Baselines

For baseline comparisons, we implemented a classical spectrogram model and two self-supervised methods, each paired with a two-layer MLP decoder (128 hidden units, ReLU) trained using Adam optimizers. Spectrograms were computed using Short Time Fourier Transform, producing 500 frequency bins × 16 time bins per trial. For **SimCLR** [34], a CNN encoder with an MLP projection head was trained on 3-second raw LFP segments (1.25 kHz) using InfoNCE loss (temperature = 0.606) with temporal masking. **BrainBERT** [35] uses a transformer encoder pretrained on spectrograms via masked token prediction. After pretraining, we fine-tune only the MLP head while freezing the encoder. All models share the same decoding architecture, isolating differences in encoder representation quality rather than decoder design.

## 5.2 Evaluation

To avoid implicitly exploiting spatial or temporal correlations between training and test data, we adopt an across-session and across-lab evaluation protocol.

**Across-session evaluation.** We split each lab's recordings by session, reserving 15% sessions for testing, 30% for validation. Models are trained on all channels and trials independently in remaining sessions, and evaluated on the held-out session. We report both balanced accuracy and macro F1 to account for class imbalance across five brain regions, and assess embedding quality via PCA of encoder representation.

**Across-lab evaluation.** To test cross-lab generalization, we train on all sessions from one lab and evaluate zero-shot on another lab with different probes and tasks. We also include one-shot transfer condition by adding a single session from the target lab to training. This setup tests each model's ability to extract anatomy-relevant features without relying on probe geometry or task structure.

# 6 Results

## 6.1 Brain Region Decoding Across Sessions

We evaluate all models on cross-session region decoding in three large-scale LFP datasets (Neuronexus, Allen, and IBL). As shown in Figure 2a, Lfp2vec achieves the highest balanced accuracy and macro F1, outperforming both the spectrogram baseline and SSL methods (SimCLR, BrainBERT), with the largest improvements on Allen and Neuronexus. Fewer performance gains are observed on IBL, likely due to its simpler classification task (fewer region classes). On the Allen dataset, confusion matrices (Figure 2b) show that Lfp2vec achieves the fewest errors across all classes, including in underrepresented subregions such as CA2 and CA3. All models perform best on CA1 and cortex, likely due to their distinctive signatures and class prevalence.

PCA projections of latent embeddings (Figure 2c) suggest that Lfp2vec produces more compact and separable clusters by region. To test whether this structure reflects neuroanatomy rather than probe-identity or session-specific artifacts, we quantitatively evaluated cluster quality using the Silhouette score (1 = perfect separation, 0 = no structure) and linear probing accuracy, as shown in Fig. 2a, right. Lfp2vec achieved a markedly higher Silhouette score for region clustering (0.576±0.026) than the spectrogram (–0.054±0.015), SimCLR (–0.062±0.050), and BrainBERT (0.146±0.026) baselines, while showing no meaningful clustering by session (–0.053±0.009) or probe identity (–0.266±0.006). Linear probing results mirrored this trend: although embeddings encode minor session and probe information, anatomical region dominates, with Lfp2vec enabling far more accurate region decoding (0.921±0.004) than baselines. Together, these findings demonstrate that Lfp2vec learns more anatomically aligned representations.

Figure 2d shows the channel-wise prediction map compared to the ground truth. Although all models capture the coarse anatomical layout, Lfp2vec's predictions more closely adhere to the known laminar organization of the hippocampus. For instance, in the example shown, it clearly delineates the CA1-DG boundary, a distinction that is less precise in the baseline models. A zoomed view (Figure 2d, right) highlights how our post-processing steps further refine these boundaries to improve localization accuracy.

## 6.2 Brain Region Decoding Across Labs

In addition to within-lab evaluation, we assess the robustness of Lfp2vec under cross-lab transfer. We initialize Lfp2vec with pretrained weights, fine tune the model on one lab, and evaluate its zero-shot prediction accuracy on another lab with different tasks and probe geometries (Figure 2e). Diagonal entries represent within-lab performance, with chance-level accuracy shown in parentheses. We observe strong zero-shot transfer between the IBL and Allen datasets, likely due to their shared use of Neuropixels probes and similar visual task structure (Figure 2e, left). In contrast, transfer from the Neuronexus dataset to others is weaker, possibly due to its distinct probe design and spontaneous recordings. When a single session from the target lab is included (one-shot transfer), accuracy improves substantially, often matching within-lab performance (Figure 2e, right), suggesting that minimal data can suffice to adapt Lfp2vec to new labs.

## 6.3 Brain Region Decoding Across Species

To test cross-species generalization, we apply Lfp2vec to the macaque Neuropixels-NHP dataset (Figure 3a) using the same across-session protocol. As shown in Figure 3b, Lfp2vec outperforms

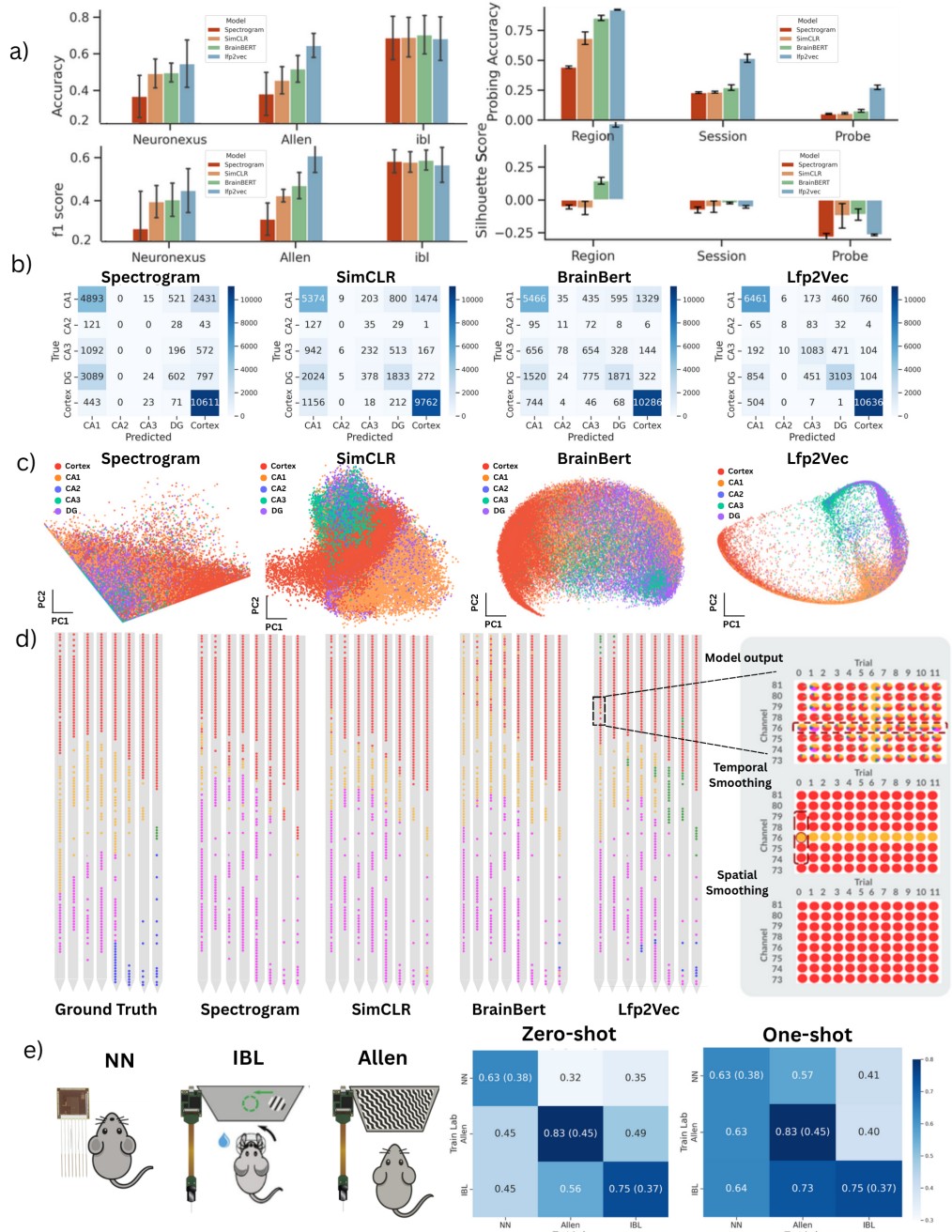

Figure 2: **Model performance comparison in region decoding across rodent datasets.** a) Balanced test accuracy and macro-F1 (left) for brain-region decoding on three mouse LFP datasets (Neuronexus, Allen, IBL). Silhouette score and linear probing accuracy (right) quantifies how well embeddings cluster by brain region, session, and probe identity in Allen sessions. b) Confusion matrices showing the brain region classification performance across all models in Allen sessions. c) PCA projections of channel embeddings for Allen sessions, colored by distinct brain regions, showing clusters by brain regions. d) Channel-wise predicted regions on a Neuronexus probe compared to ground truth. Each dot represents a region probability pie chart (right top), with temporal smoothing (right middle) and spatial smoothing (right bottom) improving prediction. e) Cross lab generalization matrix for zero-shot (middle) and one-shot (right) performance across three mice datasets (left), here high off-diagonal values indicate good generalization performance from one lab to another.

all baselines (spectrogram, SimCLR, BrainBERT), achieving the highest test accuracy and macro F1 without any species-specific adaptation. The confusion matrices (Figure 3c) reveal that Lfp2vec has the fewest misclassifications, especially between closely related regions such as SMA and M1. Latent space projections (Figure 3d) further show that Lfp2vec produces more compact and separable region-specific clusters, suggesting that its representations preserve fine anatomical distinctions. These results highlight Lfp2vec's ability to extract generalizable, anatomically meaningful features from LFP signals across species and experimental settings.

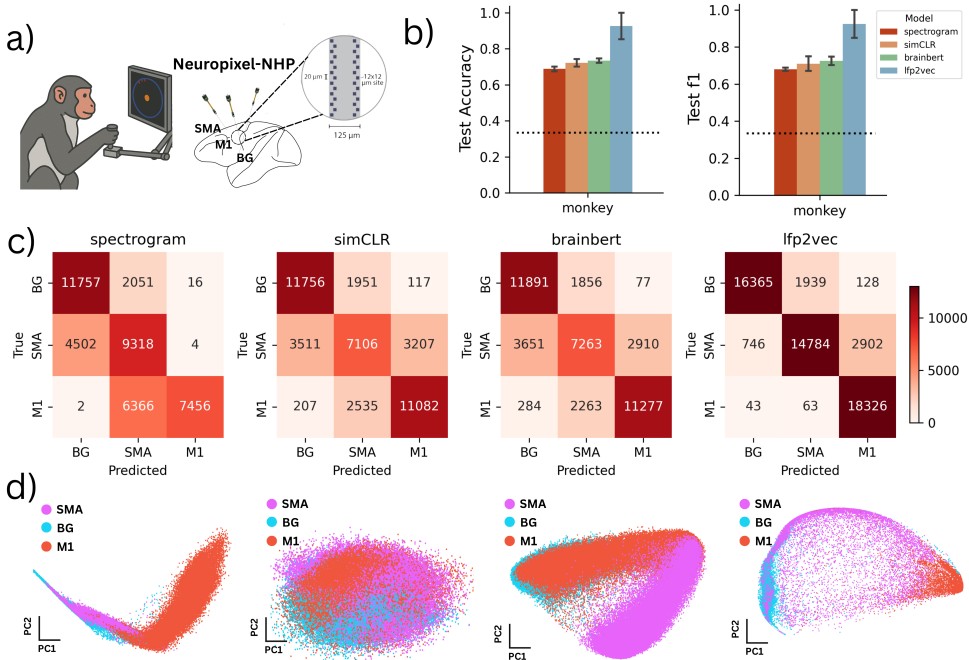

Figure 3: **Lfp2vec representations transfer across laboratories, probe geometries, and species**. a) Cross-species generalization: Lfp2vec outperforms spectrograms, SimCLR, and BrainBERT in balanced accuracy and macro-F1 for classifying SMA, M1, and BG. b) Confusion matrices show Lfp2vec achieves the highest accuracy and clearest separation across regions (BG, SMA, M1). c) PCA plots reveal Lfp2vec embeddings form distinct, region-specific clusters, generalizing beyond rodent data.

## 6.4  Disease Prediction

Besides the brain region decoding task, we evaluate Lfp2vec on a downstream task of Alzheimer's disease (AD) classification using LFP recordings from App x Psen1 transgenic mice [8]. A lightweight classifier is fine-tuned on top of pretrained Lfp2vec embeddings to distinguish AD mice from healthy controls. As shown in Figure 4a, Lfp2vec outperforms SimCLR and BrainBERT in both accuracy and F1 score. PCA projections of Lfp2vec embeddings (Figure 4b) show clear separation between diseased and healthy samples, suggesting the model captures disease-relevant neural features. To further localize abnormalities and analyze sources of error, we compute channel-level disease probabilities across trials (visualized as pie charts in Figure 4c–d), and aggregate them into region-level abnormality scores shown in the bar plots below. AD mice exhibit lower abnormality level in regions such as CA3 and DG, while healthy controls show consistently low abnormality scores across all regions. These findings highlight Lfp2vec's ability to detect and localize pathological neural signatures without supervision.

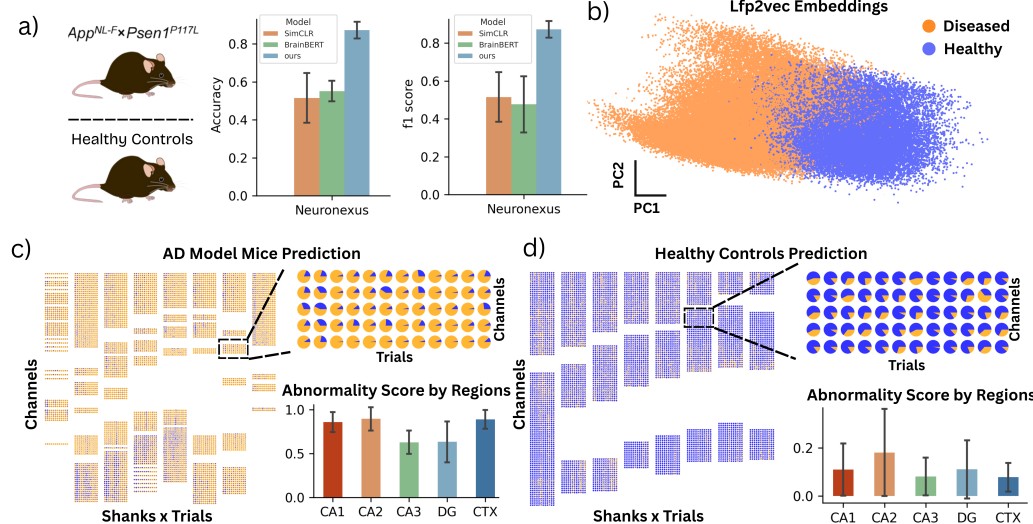

Figure 4: **Disease Prediction and Abnormality Study by Brain Regions**. a) Classification performance (accuracy and F1 score) on distinguishing Alzheimer's disease (AD) model mice (App x Psen1) from healthy controls using different self-supervised models. Lfp2vec consistently outperforms SimCLR and BrainBERT. b) PCA projection of learned Lfp2vec embeddings shows distinct clustering between diseased and healthy animals. c–d) Channel-wise predictions and region-level abnormality scores for AD model mice (c) and healthy controls (d). Each dot represents a channel's prediction across trials. Bar plots below summarize region-wise abnormality scores, showing which anatomical regions have higher deviation from normal activity. CA3 and DG show the least abnormal signals in AD mice, while abnormality scores in healthy controls remain low across all regions.

## 6.5 Ablation Study

To isolate the benefits of our learning strategy, we conducted an ablation study (Fig. 5) evaluating two key factors: the amount of self-supervised pre-training on LFP data, and the choice of model initialization. We compared models initialized with random weights against models initialized with weights from the audio-pretrained wav2vec 2.0 model. Across all datasets, continued self-supervised pre-training on LFP data consistently improves downstream decoding performance, with accuracy increasing with the number of unlabeled trials. More importantly, audio-initialized models significantly outperformed randomly initialized ones. For instance, on the Neuronexus dataset, the audio-initialized model with only a small amount of LFP pre-training (6k trials) already matches the performance of a randomly-initialized model trained on over 400k trials. This demonstrates that

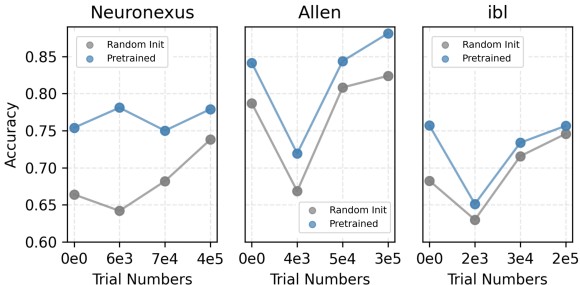

Figure 5: **Ablation Study**. Brain region decoding accuracy across three datasets (Neuronexus, Allen, and IBL) using different number of unlabeled pretraining trials for a random session. Solid lines show accuracy from models trained with self-supervised learning, initialized either randomly (gray) or with pretrained weights (blue). Pretraining consistently boosts the decoding performance and improves with more trials, demonstrating the benefit of combining SSL with pretrained models.

features learned from audio provide a strong inductive bias for modeling neural signals. The best performance is consistently achieved by combining both strategies: initializing from an audio foundation model and continuing self-supervised learning on a large corpus of in-domain LFP data.

# 7 Discussion

In this work, we presented **Lfp2vec**, a self-supervised model that learns anatomy-aware representations directly from raw local field potential (LFP) signals. Across three mouse datasets spanning probe geometries and experimental paradigms, **Lfp2vec** outperforms traditional features such as spectrograms and previous self-supervised models (SimCLR, BrainBERT) on hippocampal subregion decoding under across-session and across-lab evaluation, suggesting the representations capture fine-grained anatomical structure rather than session-specific correlations. Lightweight post-processing via temporal aggregation and spatial smoothing further improves channel-level localization. We additionally observe strong cross-domain transfer: **Lfp2vec** generalizes across laboratories and probe types (Neuropixels to Neuronexus), and extends to a non-rodent setting with improved decoding accuracy on macaque motor cortex recordings without redesigning the architecture. Finally, we demonstrate that **Lfp2vec** embeddings support a downstream Alzheimer's disease classification task, indicating that anatomically grounded representations may also capture functionally relevant biomarkers.

Despite these advances, several limitations remain. Our current study focuses on cortical and hippocampal areas due to data availability. Future work will incorporate datasets with well-labeled deep structures to test whether performance and embedding consistency persist in more heterogeneous regions. Performance is also sensitive to preprocessing (IBL destriping), suggesting that end-to-end artifact suppression or adaptive filtering could further improve robustness. Disease classification results, while promising, are based on limited mouse samples and require larger studies to assess clinical relevance in humans. Clinical translation would additionally require multi-site validation, cross-species studies, and comparison with established biomarkers. Finally, our evaluation assumes session-level splits; streaming or real-time use will likely require online adaptation.

Looking forward, we see several potential directions. Incorporating explicit spatial priors into the SSL loss could further regularize embeddings and improve cross-probe generalization. Multi-modal fusion with spiking data or imaging modalities may generate richer, interpretable representations for both neuroscience and clinical applications. From a practical standpoint, the ability to perform zero-shot localization in vivo opens the door to adaptive electrode placement and real-time feedback in closed-loop neurophysiology experiments. These methods could also be applied in chronic or clinical settings, such as locating physiological events preemptively (e.g. predict which electrodes the seizure/sharp wave ripple occur), so we can do early discharges during closed loop stimulation.

# 8 Broader Impact

Our framework for self-supervised LFP representation learning has the potential to reduce reliance on labor-intensive histology, enabling real-time electrode localization. It could support clinical applications such as adaptive deep brain stimulation and closed-loop brain–computer interfaces with targeted ultrasound neuromodulation. In safety-critical settings, incorrect localization could lead to inappropriate stimulation targets, so clinical deployment should include calibrated uncertainty estimates and human verification. For the ML community, **Lfp2vec** shows that transformer-based time series foundation models can be successfully adapted to nonstationary, noisy neural recordings, setting a new benchmark for self-supervised learning on biomedical signals. By releasing our code and pretrained weights, we aim to spur further work on domain adaptation, spatiotemporal regularization, and ethical deployment of ML methods for neural data.

## Acknowledgements

We would like to thank Saurabh Vyas for generously making their Macaque Neuropixels-NHP recordings data available. We also acknowledge Corticale Srl. for SiNAPS probes and the Radiens software for data acquisition and processing that made this research possible. This work was supported by NIH grant 1R00MN128772 and NIH grant 1R01NS113782-01A1.

Table 1: Author contributions.

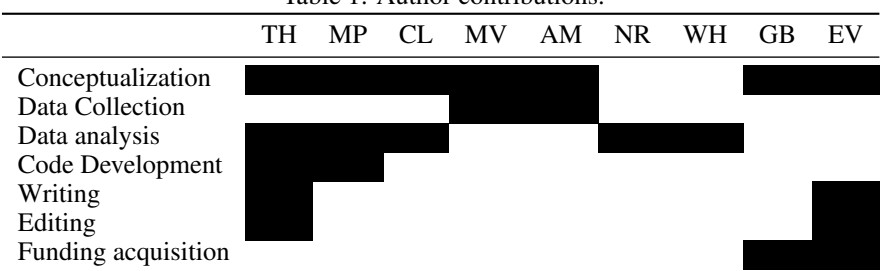

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

## Supplementary Material

## A  Brain Region Decoding Results

In addition to the Allen dataset discussed in the main text, we also present extended decoding results for the Neuronexus and IBL datasets (Figure 6). The confusion matrices show how each model performs across different brain regions. Lfp2vec consistently shows more focused diagonal patterns, indicating better region-specific accuracy across all datasets. While regions like the cortex and CA1 are reliably identified by all models, subfields such as CA2 and CA3 remain harder to distinguish. This is likely due to the sparsity of region labels and their lower biological separability. Notably, Lfp2vec maintains strong performance across all datasets. In the IBL dataset, all models perform well, possibly because of sparser labeling and the absence of more challenging classes. These findings support Lfp2vec's ability to generalize across recording platforms with varied probe geometries and signal characteristics.

## B  Baseline Settings

**SimCLR** We implement SimCLR [34] as a baseline for contrastive self-supervised learning. The input to SimCLR is a 2D spectrogram derived from the raw LFP signals. For data augmentation, we apply temporal masking, by randomly masking out segments along the time axis to encourage temporal invariance. The encoder architecture is a convolutional neural network (CNN) followed by a projection head composed of two linear layers with ReLU activation. During pretraining, we use the NT-Xent loss (normalized temperature-scaled cross-entropy loss) to maximize agreement

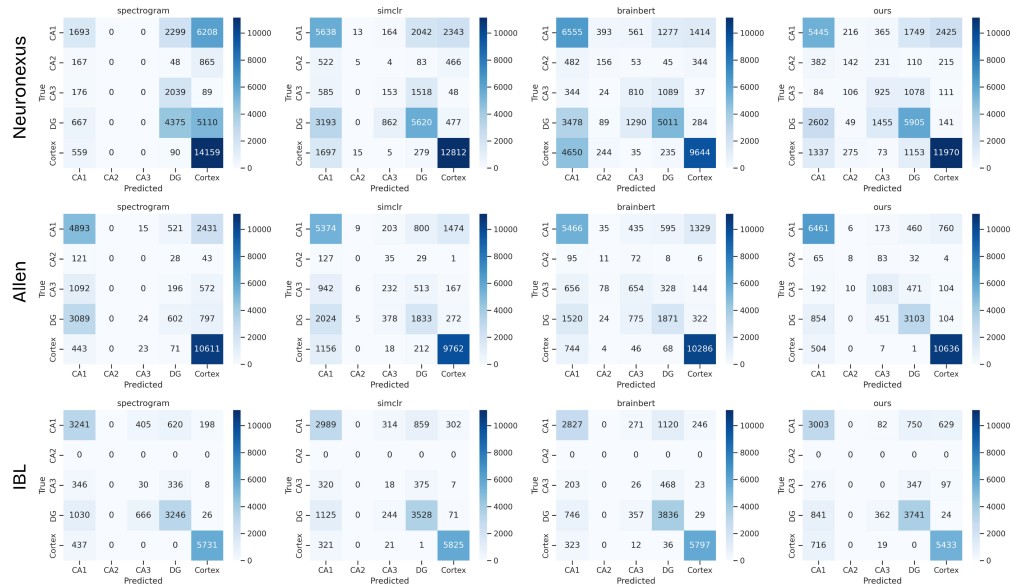

Figure 6: **Decoding performance comparisons across models and datasets**. Here we compare confusion matrix for anatomical region classification across datasets (Neuronexus, Allen, IBL) and models (spectrogram, SimCLR, BrainBERT, Lfp2vec). Each matrix shows predicted versus true region labels. Across all datasets, Lfp2vec shows more concentrated diagonal patterns. The cortex and CA1 regions are consistently well-identified across models, while hippocampal subfields such as CA2 and CA3 remain more challenging. Lfp2vec generalizes better than baselines, maintaining stable performance across datasets with diverse probe geometries and recording conditions.

between positive pairs generated from the same signal. After pretraining, we discard the projection head and fine-tune an MLP classifier on top of the encoder for brain region decoding.

**BrainBERT** We adopt BrainBERT following the implementation in [35], which adapts a transformer architecture to 2D spectrograms from electrophysiology signal. The model is trained from scratch on our dataset using masked token prediction as the pretraining task. After pretraining, the transformer encoder is frozen, and we train an MLP classification head for downstream decoding. The MLP consists of two fully connected layers with ReLU activation and dropout. We use Cross Entropy Loss for classification and perform Bayesian optimization over the learning rate, dropout rate, and hidden size of the MLP to select hyperparameters. This two-stage training mirrors the original BrainBERT protocol while allowing adaptation to our LFP data.

## C Hyperparameters Settings

**Training Schedule.** We use separate training schedules for pretraining and fine-tuning in Lfp2vec. During self-supervised pretraining, the encoder is optimized using the AdamW optimizer with a learning rate of 1e-5, a batch size of 32, and trained for 50 epochs. For fine-tuning, we use a learning rate of 3e-5, keep the batch size at 32, and train for 10 epochs. We apply gradient accumulation over 4 steps to simulate a larger effective batch size, and use a linear warmup schedule over 10% of the total training steps. This specific set of hyperparameters are suitable for Allen dataset, and may require further hyperparameter tuning for novel datasets.

**Model Architecture.** The model architecture consists of four main components: a convolutional encoder, a product quantizer, a Transformer module, and an MLP classifier. The encoder is a 7-layer 1D convolutional stack with kernel sizes of (10, 3, 3, 3, 3, 2, 2) and strides of (5, 2, 2, 2, 2, 2, 2), each followed by GELU activation and LayerNorm.

Table 2: Hyperparameter Settings for Lfp2vec

| Stage | Hyperparameter | Value(s) | Description |
|---|---|---|---|
| Pretraining | Learning Rate | $1 \times 10^{-5}$ | Step size for encoder during self-supervised learning |
| | Batch Size | 32 | Number of samples per training batch |
| | Epochs | 50 | Number of training iterations |
| | Optimizer | AdamW | Optimization algorithm |
| Fine-tuning | Learning Rate | $3 \times 10^{-5}$ | Step size for encoder during fine tuning |
| | Batch Size | 32 | Number of samples per training batch |
| | Epochs | 10 | Number of fine-tuning iterations |
| | Optimizer | AdamW | Optimization algorithm |
| | Gradient Accumulation | 4 | Batches per optimizer step |
| | Warmup Ratio | 0.1 | Fraction of steps for learning rate warmup |

This produces a 512-dimensional latent representation at each time step. The quantizer discretizes these features using Gumbel-softmax sampling. A diversity loss is applied to encourage full usage of the codebooks. The Transformer module contains 12 layers with 12 attention heads, a hidden size of 768, a feedforward size of 3072, and a dropout rate of 0.1. Finally, the classifier is a two-layer MLP with 256 hidden units and ReLU activation, mapping the output to one of five brain region classes.

Table 3: Model Hyperparameters for Lfp2vec

| Component | Parameter | Value |
|---|---|---|
| **Encoder** | # Layers | 7 |
| | Kernel Sizes | (10, 3, 3, 3, 3, 2, 2) |
| | Strides | (5, 2, 2, 2, 2, 2, 2) |
| | Activation | GELU |
| | Normalization | LayerNorm per layer |
| | Output Dim | 512 |
| **Quantizer** | Codebooks | $2 \times 320$ |
| | Total Dimension | 640 |
| | Sampling Method | Gumbel-softmax |
| | Auxiliary Losses | Diversity Loss |
| **Transformer** | # Layers | 12 |
| | # Attention Heads | 12 |
| | Hidden Size | 768 |
| | Feedforward Size | 3072 |
| | Dropout | 0.1 |
| **Classifier** | Hidden Units | 256 |
| | Activation | ReLU |
| | Output Classes | 5 (brain regions) |

# D  Post-processing

We post-process the model's per-timepoint predictions $p_{t,c} = p(y = c \mid x_t)$ by averaging class probabilities across a temporal window. This acts as a lightweight denoising step that suppresses local fluctuations in predictions. For each channel $c$, we compute a smoothed class score:

$$\bar{p}_c = \frac{1}{|\mathcal{W}_t|}\Big(\sum_{t' \in \mathcal{W}_t} p_{t',c}\pi_c\Big),$$

where $\mathcal{W}_t$ denotes a symmetric window around $t$. This temporal smoothing preserves softmax semantics and can be interpreted as a local belief update under the assumption of short-term temporal consistency. And we optionally incorporate a class prior $\pi_c$ to bias predictions toward frequent classes. $\pi_c$ may be uniform or estimated empirically from the training distribution. The final prediction is obtained by taking the most probable class:

$$\hat{y}_t = \arg\max_c \tilde{p}_{t,c}.$$

This smoothing strategy increases temporal coherence without introducing model parameters or transition assumptions. For spatial smoothing, we apply majority voting across neighboring channels within each timepoint. Specifically, each prediction $\hat{y}_{t,c}$ is replaced with the most frequent label among a fixed spatial neighborhood $\mathcal{N}_c$ around channel $c$. This encourages spatial contiguity and reduces anatomically implausible local fluctuations.

The effectiveness of these post-processing strategies is summarized in Figure 7. The temporal smoothing ensures that each channel receives a single, consistent anatomical label across trials. And the spatial smoothing ensures that there are no discontinuities in channel prediction. As shown in panels b, d, and f, this significantly reduces noise-driven fluctuations in the raw predictions and produces cleaner anatomical maps, particularly in deeper structures where variability across trials is most pronounced. The accuracy plots (a, c, e) confirm that this approach yields measurable improvements, especially for Lfp2vec, which produces rich but sometimes inconsistent predictions. These findings highlight that post-processing does not merely boost metrics, but selectively compensates for dataset-specific and region-specific noise patterns inherent in neural data.

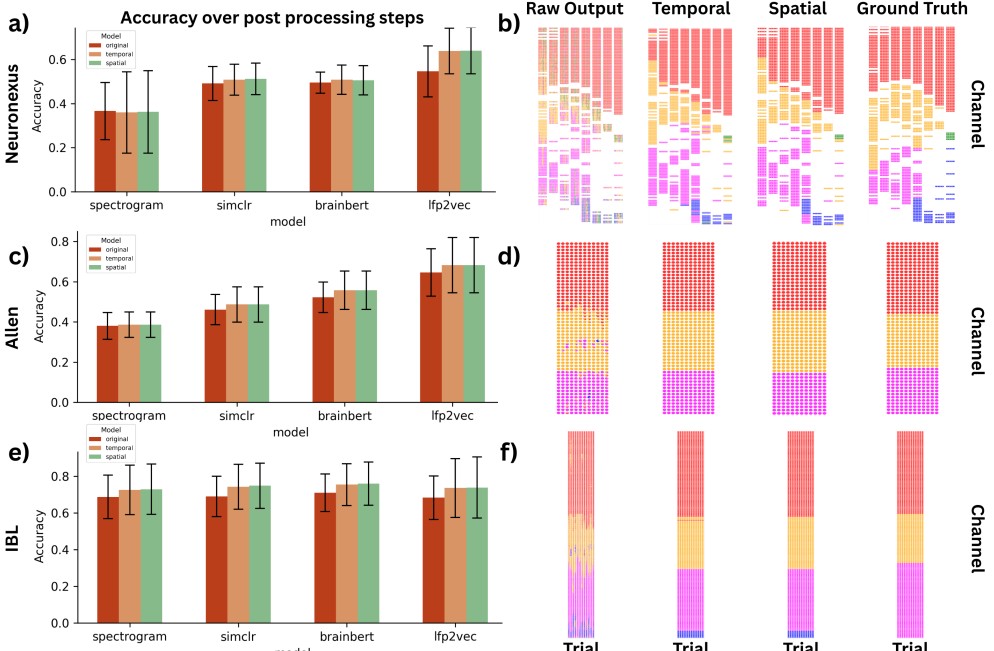

Figure 7: **Effect of post-processing on anatomical decoding across datasets and models.**. a, c, e) Classification accuracy before and after post-processing (temporal smoothing, spatial smoothing) across datasets (Neuronexus, Allen, IBL) and models. Accuracy consistently improves or remains stable after post-processing, with the largest gains seen for Lfp2vec. b, d, f) Visualization of predicted anatomical labels across channels and trials for an example session, before and after post-processing, compared to ground truth. Raw predictions exhibit spatial and temporal discontinuity, particularly in deeper regions. Temporal and spatial smoothing align predictions more closely with anatomical boundaries, reducing local inconsistencies and increasing interpretability.

## NeurIPS Paper Checklist

The checklist is designed to encourage best practices for responsible machine learning research, addressing issues of reproducibility, transparency, research ethics, and societal impact. Do not remove the checklist: **The papers not including the checklist will be desk rejected.** The checklist should follow the references and follow the (optional) supplemental material. The checklist does NOT count towards the page limit.

Please read the checklist guidelines carefully for information on how to answer these questions. For each question in the checklist:

- You should answer [Yes] , [No] , or [NA] .

- [NA] means either that the question is Not Applicable for that particular paper or the relevant information is Not Available.

- Please provide a short (1–2 sentence) justification right after your answer (even for NA).

**The checklist answers are an integral part of your paper submission.** They are visible to the reviewers, area chairs, senior area chairs, and ethics reviewers. You will be asked to also include it (after eventual revisions) with the final version of your paper, and its final version will be published with the paper.

The reviewers of your paper will be asked to use the checklist as one of the factors in their evaluation. While "[Yes] " is generally preferable to "[No] ", it is perfectly acceptable to answer "[No] " provided a proper justification is given (e.g., "error bars are not reported because it would be too computationally expensive" or "we were unable to find the license for the dataset we used"). In general, answering "[No] " or "[NA] " is not grounds for rejection. While the questions are phrased in a binary way, we

acknowledge that the true answer is often more nuanced, so please just use your best judgment and write a justification to elaborate. All supporting evidence can appear either in the main paper or the supplemental material, provided in appendix. If you answer [Yes] to a question, in the justification please point to the section(s) where related material for the question can be found.

IMPORTANT, please:

- **Delete this instruction block, but keep the section heading "NeurIPS Paper Checklist",**
- **Keep the checklist subsection headings, questions/answers and guidelines below.**
- **Do not modify the questions and only use the provided macros for your answers.**

