# OpenReview forum: "Self supervised learning for in vivo localization of microelectrode arrays using raw local field potential"
_NeurIPS.cc/2025/Conference — NeurIPS 2025 poster_

### Official Review · Reviewer_KDGD · 2025-06-24

**Clarity:** 4
**Significance:** 3
**Originality:** 2
**Rating:** 5
**Confidence:** 4

**Summary:**

The paper introduces Lfp2Vec, a contrastive, masked SSL transformer method that utilizes Wav2vec-like training to learn neural embeddings from LFP data.

The embeddings exhibited high performance on a variety of different downstream settings, including within-session, cross-lab, and even cross-species, as well as on classifying alzheimer's disease.

**Questions:**

- Where do you foresee these methods being applied?
- How do you think the performance would be affected if more areas were imaged?
- is there any analysis on the dynamics of the vector quantizer? Do the learnt representations also make sense? Would you expect them to?

**Ethical Concerns:**

["NO or VERY MINOR ethics concerns only"]

**Final Justification:**

The paper is well presented, impactful, and is broadly applicable to important neuroscientific research.

**Limitations:**

- Although method has shown superior performance on a good amount of tasks already, it could be tried on more extensive spans of areas (deeper cells) to see if this performance chasm remains.

**Paper Formatting Concerns:**

- Table 1. Too much white space, perhaps wrapping could help.

**Quality:**

4

**Strengths And Weaknesses:**

**Strengths**
- Improved results across the board.
- Variety of tasks and settings.
- Tested on both cortex and hippocampus regions.
- Method is generalizable, and makes intuitive sense. Has been used in other neural tasks (CEBRA, Neuroformer, etc).
- Informative figures, paper is in general well-presented and written.

**Weaknesses**
- Similar methods have already been attempted.
- Sparse reporting. Exact accuracies for the tasks are missing, including confidence intervals and significance testing.
- It would be desirable to also show results on deeper areas if possible.
- The codebook dynamics for the learnt vector quantizer were left unexplored. This would be an interesting area of exploration.

---

> ### Author Rebuttal · Authors · 2025-07-31
>
> We thank the reviewer for their positive feedback on the robustness and generalizability of our model across tasks. The reviewer raised three main concerns: model novelty, incomplete reporting, and lack of codebook interpretation. We address these concerns by clarifying the model's novelty, adding error bars to the results, and providing visual analysis of the codebook vectors.
>
> **Weaknesses**:
> - While prior work exists, our approach is the first to apply SSL directly on raw LFPs for anatomical decoding across species and labs.
> - To address the sparse reporting, we replaced Table 1 with Supplement Figure 6, and added Figure 3 with Supplement Figure 5. These additions made sure that all accuracy values are reported with error bars.
> - We agreed that deeper region results are valuable. This is our important next step to extend this method to thalamus, and whole brain regions.
> - We thank the reviewer for this suggestion. We applied t-SNE to analyze the quantized vectors in the codebook and included it in the revised manuscript.
>
> **Questions**:
> - Following this suggestion, we expanded the broader impact section: “these methods could be applied in physiological experiments to provide real time feedback, or in chronic or clinical settings, or locating physiological events preemptively (e.g. predict which electrodes the seizure/sharp wave ripple occur), so we can do early discharges during closed loop stimulation. “
> - With broader spatial coverage, we expect performance to vary by region. Potential factors affecting the performance are signal quality, number of labels (sparse labels often give poor performance), and if there exist signature biomarkers for sublayers.
> - We visualized the codebook and will include them in the revised manuscript.
>
>
> **Limitations**:
> - We agree that testing in deeper brain regions (e.g.,thalamus, basal ganglia) is important for assessing generalizability. Our current study focused on cortical and hippocampus areas due to data availability. As a next step, we plan to incorporate datasets with well-labeled deep structures to evaluate whether performance and embedding consistency persist in more heterogeneous regions.
>
> **Formatting**:
> - We removed Table 1 and replaced it with Supplement Figure 6

---

### Official Review · Reviewer_jdpB · 2025-07-01

**Clarity:** 4
**Significance:** 3
**Originality:** 2
**Rating:** 5
**Confidence:** 4

**Summary:**

This work presents an SSL architecture and training strategy heavily inspired by wav2vec. It employs 1D convolution encoding layers, a discrete codebook, and Transformer for masked reconstruction training. Embeddings are straightforwardly used with finetuned classification layers, whether for region or disease decoding. Model performance compared to existing baselines appear competitive, with potential for across-lab and across-species generalization.

**Questions:**

1. how large is within-session variability of embeddings, and therefore, the decoded brain region?
2. Similarly, how does representation change according to task?
3. line 166: please clarify how a 50Hz sequence post-convolution preserves fast oscillatory dynamics, which the various forms of gamma in the hippocampus can have frequencies much higher than the Nyquist rate of 25Hz here.
4. I’m confused how the codebook dimensions add up to 512? Or is it 4 different codebooks of various sizes, all with a code dimensionality of 512?
5. I think a linear probing experiment would be informative to assess the learned representations, instead of having to finetune an MLP, Transformer, and the code quantizer.
6. Have the authors tried other masking strategies, aside from full time index masking, assuming that means masking out tokens at individual timepoints in their entirety? What about random masking over time and dimensions?
7. Please clarify how the context and distractor tokens are chosen, based on temporal distance?
8. Is the 5-NN majority vote procedure iterative, or simply done in one sweep based on the original model prediction?
9. For how long is it safe to assume that channels sit in fixed anatomical locations (and layers) across all trials (windows) of one recording? Does one observe consistent prediction shifts over time for a long recording? (referencing line 199)
10. Are the post-processing steps also applied to the other baseline models in Figure 2a & b? It is implied in the caption for d, but it’s unclear whether that applies to the previous results in the figure.
11. Is there a quantification / ablation study of the impact of postprocessing?
12. Apologies if I missed this, but how exactly is the unlabeled data for SSL and labeled data for fine-tuning split? Line 225 states zero-shot prediction is tested on test sessions. So are pretraining and finetuning data the same trials in the training set, just used without and then with labels?
13. Just to clarify: the model operates entirely on 1D time series, on a basis of 3s snippets of individual channels? This is how it generalizes to other recording setups without reconfiguring the convolution encoders?
14. why did one-shot finetuned performance decrease, e.g., for Allen to IBL transfer?
15. It’s difficult to evaluate the consistency of higher performance in Figure 3 and Table 1 without error bars, though the absolute performance is nevertheless impressive.

**Ethical Concerns:**

["NO or VERY MINOR ethics concerns only"]

**Final Justification:**

Technically sound paper, reasonable evaluations, and an adequate ML solution to an important and practical neuroscience problem.

**Limitations:**

Reasonable discussions of limitations in the proposed method given space constraint.

**Quality:**

3

**Strengths And Weaknesses:**

Strengths
- tackles an important problem of using LFPs, which is much more readily available and easier to standardize, for anatomical localization. This is a very feasible use case so a compelling solution would be an important contribution.
- really well written manuscript and I enjoyed reading it; the text is informative but concise, and adequately covers literature in both neuroscience and ML.
- architecture and training strategy mirrors wav2vec2.0, is simply, elegant, and sensible
- the post processing strategy is also interesting, and appears to boost performance
- evaluation experiments appear to be rigorously setup (e.g., across-session splits), and the results presented seem convincing and is well-visualized
- code is already available!

Weaknesses
- the model boasts prediction of detailed anatomical information, but all quantitative performance shown were for multi-class region classification. It should be straight-forward to extend to regression onto a continuous measure? It would be very interesting to apply the proposed method for really fine-graned depth localization use cases.
- while the evaluation results are interesting and impressive in most cases, there remains some concerns regarding the setup and lack of variance reporting.
- given that LFP signals change drastically as a function of task and behavioral state, I find it surprising that there is no evaluation of embedding or prediction consistency as a function of time / state
- there are some overlap in text in section 5.2
- the visualization in Fig 3a is a bit confusing, as one expects it to be a confusion matrix with low off-diagonal values, but it’s rather cross-dataset performance, such that off-diagonal values are ideally high.

---

> ### Author Rebuttal · Authors · 2025-07-31
>
> We thank the reviewer for their positive feedback on the robustness of our method and its practical significance to neurophysiology. The reviewer raised three main concerns: the temporal stability of predictions and embeddings, the absence of variance reporting, and the limited explanation of design choices. We address these concerns by conducting additional experiments to analyze prediction and embedding consistency, updating our results to include error bars, and providing further clarification on our design choices.
>
> **Weaknesses**:
>
> “[Can we extend]...region classification…to a continuous measure? “
>
> - We can extend this approach to the Allen dataset, which includes continuous anatomical labels from the Allen Brain Atlas Common Coordinate Framework (CCF), providing x, y, z coordinates for each channel. By replacing the classification objective with a regression objective (e.g., MSE loss), the model can be fine-tuned to predict spatial coordinates directly. However, the IBL and Neuronexus datasets do not provide such continuous labels, so we focused on region classification in this study.
>
> “concerns regarding the setup and lack of variance reporting”
>
> - We thank the reviewer for raising this point. We have removed Table 1 and replaced it with Supplementary Figure 6, which includes error bars to reflect variance in disease classification across folds of test sessions. We also added error bars to Figure 3 in the revised manuscript.
>
> “no evaluation of embedding or prediction consistency as a function of time / state”
>
> - We thank the reviewer for raising this point. To address this, we added additional analysis of prediction consistency over time/trials. Here the first row is the mean variance of prediction probabilities across trials, averaged across sessions, second row is the mean cosine similarity for embeddings across trials, averaged across sessions.
>
> |                         | **Spectrogram**     | **SimCLR**          | **BrainBERT**       | **lfp2vec**         |
> |-------------------------|---------------------|---------------------|---------------------|---------------------|
> | **Prediction Variance** | 0.004 ± 0.002       | 0.012 ± 0.007       | 0.018 ± 0.005       | 0.018 ± 0.007       |
> | **Embedding Cosine Similarity** | 0.897 ± 0.151 | 0.551 ± 0.212       | 0.863 ± 0.051       | 0.864 ± 0.036       |
>
>
> - The small prediction variance indicates that our model, along with other baselines, produces consistent predictions across trials. Embedding similarity, measured as the average cosine similarity between normalized embedding vectors across trials, is also high in our model, suggesting that the learned representations are stable over time.
>
>
> “visualization in Fig 3a is confusing as one expects confusion matrix…overlap in text in Section 5.2”
>
> - We agree that current Fig 3a may resemble confusion matrices and lead to misunderstanding. To clarify, we added the following text in Fig 3a caption: “Cross lab generalization matrix for zero-shot and one-shot performance, here high off-diagonal values indicate good generalization performance from one lab to another”. We also cut Section 5.2 to avoid redundancy.
>
>
> **Questions**:
>
> - To evaluate within session variability, we compute embedding and prediction variance  across trials, as shown in Weakness # 3. Results showed that our model produces stable representation and consistent prediction with low variability across trials. Also, our model does not explicitly leverage task structure, and achieves accurate region decoding in a task agnostic manner.
> - We agree that signals above the Nyquist frequency cannot be faithfully reconstructed after downsampling. But empirically, we observe that the embeddings retain some high-frequency information, suggesting that the convolutional layers do not entirely filter out high-frequency components. One possible explanation is that high-frequency bursts may leave detectable traces in the lower-frequency structure, which the model may still exploit. To further investigate this, we plan to visualize the learned convolutional filters to understand which frequency components they emphasize or suppress.
> - We apologize for the oversight. This was due to changes across multiple configurations during tuning. The final model uses two notebooks, each with a size of 320, resulting in a total dimension of 640.
> - We agree that linear probing would offer a clearer view of the representations' quality and will include it in future analysis.
> - We thank the reviewer for this insightful point. Our model currently uses random time masking within each trial. Since inputs are 1D signals and each trial-channel pair is treated independently, no inter-trial structure is explicitly modeled.
> - In response to the suggestion, we clarified this in Section 4.3: “Context tokens are the transformer outputs at masked time steps; target tokens are quantized vectors at the same time step; distractors are quantized vectors sampled from all other time steps in the same batch.”
> - 5-NN majority vote is applied in a single pass. While an iterative approach could be explored, it may cause over-smoothing. In practice, one iteration is usually sufficient to correct spontaneous errors.
> - To ensure recording stability, we rely on high-quality data from collaborators and preprocessing tools such as DREDGE [1] to correct for motion artifacts. As an additional consistency check, we analyzed prediction variance across time in Weakness #3 and confirmed our model's robustness.
> 	[1] DREDGE, Windolf et al. , Nature Methods 2025
> - We apologize for the confusion in Figure 2d and have revised its caption as follows:: Figure 2d (left) channel-wise prediction before post processing, Figure 2d (right) example of post processing applied on lfp2vec outputs.
> - To quantify the impact of post-processing, we added Supplementary Figure 7. Post-processing improves performance across all models, with the largest gain observed for LFP2vec.
> - We clarified Section 5.2 to specify: “We use the same unlabeled data for SSL training and the same data with labels for supervised fine-tuning.”
> - If the sampling frequency (1250 Hz) and trial length (3 s) are kept the same, the model should generalize across recording setups without reconfiguration.
> - When transferring from Allen to IBL, the model is pretrained on a larger, diverse dataset. In comparison, one labeled IBL session has fewer channels and less anatomical diversity. If we fine tune on the small, biased sample, it may skew the decision boundary and bias the model away from the more generalizable Allen representations.
> - To address the consistency issue, we removed Table 1 and replaced it with Supplement Figure 6 with error bars. We also added error bars for Figure 3, and the confusion matrix can be found in Supplement Figure 5.

---

> ### Comment · Reviewer_jdpB · 2025-08-04
>
> I thank the authors for their detailed response. Their clarifications, along with the additional temporal stability analysis, made the paper stronger imo, and I therefore increase my score. Still, I would have liked to see results on the xyz coordinate prediction, instead of simply stating that it is possible given the Allen dataset, as this would have been a much more compelling result (or, to see that it can be further improved).

---

### Official Review · Reviewer_4cyY · 2025-07-02

**Clarity:** 3
**Significance:** 3
**Originality:** 3
**Rating:** 5
**Confidence:** 4

**Summary:**

•	This paper presents a method for predicting the location, i.e., the region where an electrode is implanted, based on the LFP signal recorded by that electrode. The method is based on wav2vec. Using self-supervision, a model is pre-trained to learn a contrastive objective. The authors then demonstrate that information about electrode location can be decoded from the resulting representations. They also show that the representations are useful for other downstream tasks.

**Questions:**

•	Line 191: What is $\mathcal{K}$? Is it the set of distractors unioned with the true codebook word?
•	Section 4.3: Earlier, on line 171, commitment and diversity losses were mentioned. Are they simply added to the loss on line 191?
•	Figure 2d: What does “temporal smoothing” refer to? I can’t find it in the text.
•	Figure 2d: The caption says “Channel-wise localization accuracy before and after…post-processing”. This makes sense for the right side of Fig 2d. But for the left-hand side: are the probes shown before or after post-processing?  What does “rescues baseline models” mean in the context of this figure?

Small things
•	Figure 1 Task 1: This is a little hard for me to interpret on the first reading. What do the columns correspond with? Are they probes? What are the colors? Are they regions?
•	Section 4.1: There are many abbreviations here: “LFP”, “IBL”, “DC”, “ADC”. I think “ADC” and “IBL” should be spelled out at least once.
•	Section 4.3: The description of the method is sparse. I assume this is because most of it is similar to the objective in wav2vec2.0. It could be worth mentioning that.
•	Figure 3b: The bar chart label says “wav2vec”

**Ethical Concerns:**

["NO or VERY MINOR ethics concerns only"]

**Final Justification:**

The thorough response from the authors and their answers to the questions I raised have convinced me that this manuscript is suitable for publication.

**Limitations:**

Yes

**Paper Formatting Concerns:**

No major formatting issues

**Quality:**

3

**Strengths And Weaknesses:**

Pros
•	The task of anatomical source localization seems like a novel application for a self-supervised approach
•	The post-processing seems novel and highlights the usefulness of having priors from neuroscience, i.e., knowing the relative size of regions. I imagine the post-processing could be made even more opinionated by injecting knowledge about which regions are proximal to which other regions.
•	Presentation of methods is well-written and clear.
•	Results are shown for ambitious cross-lab and cross-species settings
Cons
•	The wav2vec2.0 architecture and approach itself are not novel. And other approaches have used similar wav2vec2.0 based approaches for LFP, e.g., BENDR https://arxiv.org/abs/2101.12037
•	The paper proposes using a deep-learning approach to predict the anatomical location of electrodes. I have two doubts about the utility of such an approach. First, predicting a region label does not give the same information as co-registration with a pre-op MRI. The region label is much coarser. I do grant that this won’t matter for all analyses. Second, this method has much better performance in the cross-session and one-shot settings, as opposed to the cross-lab and zero-shot settings (Fig 1a and Fig 3). This is a little disappointing because, in the cross-session or one-shot setting, ground-truth labels for the location are already available, and prediction isn’t needed. Or am I misunderstanding something?

---

> ### Author Rebuttal · Authors · 2025-07-31
>
> We thank the reviewer for their positive feedback on our novel application of self-supervised learning to source localization, and effective usage of postprocessing as a neuroscience prior . The reviewer raised three main concerns: model novelty, the value of anatomical decoding, and unclear cross-lab evaluation. We address these by clarifying differences from prior methods like BENDR, highlighting the role of anatomical decoding when MRI is unavailable, and clarifying the cross-lab evaluation setup.
>
>
> **Weakness**:
>
> “Other approaches have used similar wav2vec2.0 based approaches for LFP, e.g., BENDR”
> - We thank the reviewer for highlighting relevant prior work such as BENDR. In the related work section, we have added the following: “Other models like BENDR [1], BrainBERT [2], EEG2REP [3], EEG-BERT [4] and related intracranial models operate on broadband signals to decode speech, behavioral, or cognitive states. While effective, these models are not designed for anatomical localization and do not incorporate anatomical priors or address cross-lab/species generalization in invasive electrophysiology. In contrast, our model operates directly on raw LFP signals and targets in vivo anatomical decoding, bridging neuroanatomy and self-supervised learning.”
>
> [1] Kostas et al., Frontier in Human Neuroscience, 2021
> [2] Wang et al., ICLR, 2023
> [3] Foumani et al., KDD, 2024
> [4] Wang et al., biorxiv, 2024
>
> “Predicting a region label does not give the same information as co-registration with a pre-op MRI”
> - We thank the reviewer for raising this concern. We have revised the introduction to clarify that, unlike raw LFP traces, pre-op/post-op MRI is not always available in chronic studies or during surgery. Even when available, MRI lacks the spatial resolution needed to distinguish fine subregion structures such as CA1, CA2, CA3, DG in the hippocampus.
>
> “better performance in cross-session and one-shot …[than] the cross-lab and zero-shot. [But] in cross-session and one-shot, ground-truth labels … are already available”
> - We thank the reviewer for the comment and have clarified this in the revised manuscript. All evaluations are zero-shot on new insertions or new subjects, with no ground truth labels at inference time. Cross-lab zero-shot involves no examples from the new lab and is the most challenging task. One shot uses a single labeled session but is still evaluated on unseen insertions or subjects in the new lab. What our results show is that, (1) if a lab uses similar probes and task setup as Allen/IBL, this model generalizes well in zero-shot; (2) if the task/probe is new, one labeled session is enough to enable generalization to the future subjects using the same task/probe. With more diverse lab data in the future, the next step is good zero-shot transfer to new labs.
>
>
> **Questions**
>
> - Yes, we revised line 191 to clarify this. Specifically, we now add: “where K(m) denotes the set of distractors unioned with the true quantized vector q_m”
> - Yes, we clarified this in line 171 with the following addition: “This diversity loss is added to contrastive loss, weighted by a hyperparameter lambda that controls its strength”.
> - “Temporal Smoothing” in Figure 2d refers to the method described in Section 4.4. It involves computing a weighted average of label probabilities over a time window.
> - We revised the caption in Figure 2d to clarify it: Figure 2d (left) channel-wise prediction before post processing, Figure 2d (right) post processing applied on lfp2vec outputs.
> - We added more labels and legends in Figure 1 Task 1 and clarified this in the caption. “Dot plot shows predicted region probabilities per trial. Columns correspond to trials, rows to channels, and colors represent brain regions (matching the legend in middle panel).
> - We revised the text to define all abbreviations at first mention: IBL is the dataset collected by International Brain Lab, LFP is Local - - Field Potential signal, DC is direct current, ADC is Analog-to-Digital Converter in Neuropixel probes.
> - Thank you for pointing this out. We now include in Section 4.3, “The objective closely follows the wav2vec 2.0 contrastive learning framework.”
> - Apologies for the labeling error, we corrected it to “lfp2vec” in the revised Figure 3b.

---

> > ### Comment · Reviewer_4cyY · 2025-08-05
> >
> > The authors have answered all my questions and addressed my concerns, so I am increasing the rating to 5.

---

### Official Review · Reviewer_vF5b · 2025-07-02

**Clarity:** 3
**Significance:** 3
**Originality:** 2
**Rating:** 5
**Confidence:** 4

**Summary:**

This paper introduces Lfp2Vec, a self-supervised "foundation model" that learns embeddings directly from raw local field potential (LFP) traces recorded by high-density probes. Adapting the wav2vec 2.0 masked-prediction framework to electrophysiology, the model uses a 1D conv front-end plus a 12-layer Transformer to output 512D embeddings without any anatomical labels. The authors pre-train on three large mouse datasets collected in different laboratories (Allen Visual Coding, IBL Repro-Ephys, and 1024-channel Neuronexus SiNAPs recordings) and evaluate on four tasks. Across all datasets Lfp2Vec outperforms handcrafted spectrogram features, SimCLR on raw LFP, and the spectrogram-based BrainBERT, achieving accurate zero-shot region predictions and forming anatomically coherent embedding clusters. Minimal post-processing further boosts per-channel localization. Thus raw LFPs contain generalizable anatomical/functional cues, and Lfp2Vec could aid real-time probe placement and biomarker discovery.

**Questions:**

* You show a single zero-shot result on one macaque dataset. Could you (a) add a within-species baseline (macaque-only training) to quantify the transfer gap, and/or (b) report channel-wise confusion to reveal which regions fail?
* Region clusters might reflect probe-type or amplifier signatures rather than neuroanatomy. Please test a domain-adversarial experiment (e.g., predict probe ID from embeddings) or provide a confusion table stratified by probe model after adversarial training
* Can you visualize Transformer attention heads or perform a linear regression from embedding dimensions to canonical LFP bands (delta, theta, ripple)? Even qualitative insight would help users trust and adopt the model.

**Ethical Concerns:**

["NO or VERY MINOR ethics concerns only"]

**Final Justification:**

The rebuttal has alleviated (minor) concerns about cross-species evaluation, probe-type bias in clustering, and biological insight. I will maintain my positive score

**Limitations:**

* The Alzheimer’s-model demo is preliminary. Clarify that the model is *not* validated for diagnostic use and discuss pathways (multi-site trials, regulatory approval) needed before clinical deployment.
* The cross-species results are anecdotal; please state explicitly that performance on primate or human iEEG/LFP remains unverified and may require fine-tuning.
* While rodent LFP poses minimal privacy risk, future human recordings could be re-identifiable. Discuss anonymization and consent practices that would be needed.

**Paper Formatting Concerns:**

Acronyms (e.g., SiNAPs, RPLR) appear before definition

**Quality:**

3

**Strengths And Weaknesses:**

This is a technically solid, clearly written paper that convincingly shows self-supervised masked prediction on raw LFP yields embeddings useful for anatomical localization and several downstream tasks, with promising cross-lab generalization. Originality is moderate but practical significance for neurophysiology is high.

QUALITY

Strengths:
* The study combines 3 high-density mouse datasets from different labs
* straightforward but effective adaptation of wav2vec-2.0 masking to 1D LFP, with ablations on spectogram input, SimCLR raw-waveform, and BrainBERT baselines
* Rigorous benchmark evaluation (cross-lab transfer, downstream tasks, etc.)
* Code link and dataset identifiers are provided

Weaknesses:
* Cross-species test is thin, with only one macaque dataset and a single motor task
* Embedding dimensions and attention maps are not analyzed; biological insight is limited to UMAP plots
* Probe-type or amplifier signatures could drive region clustering

CLARITY

Strengths:
* Figures crisply show architecture, masking scheme, and localization heat-maps
* Results tables separate in-lab, cross-lab, and cross-species settings
* Writing is concise; methodological flow is easy to follow

Weaknesses:
* Key preprocessing steps (filter bands, re-referencing) are relegated to the supplement
* Statistical significance notation is inconsistent across plots

SIGNIFICANCE

Strengths:
* Accurate, label-free channel localization could aid real-time probe placement and large-scale annotation pipelines
* Demonstrates that raw LFP encodes transferrable anatomical/functional signatures, a useful message for systems and clinical neurophysiology communities

Weaknesses:
* Broader AI relevance is moderate: the technique is a straightforward domain transfer of wav2vec rather than a new representation-learning principle

ORIGINALITY

Strengths:
* First work (to my knowledge) to apply masked predictive coding à la wav2vec to high-density LFP
* Cross-lab foundation-model framing is novel within electrophysiology

Weaknesses:
* Conceptually similar to prior BrainBERT/EEG-BERT work; innovation lies mainly in domain shift rather than algorithmic novelty
* No new masking strategy or architecture variant specific to bio-signals is proposed

---

> ### Author Rebuttal · Authors · 2025-07-31
>
> We thank the reviewer for the thoughtful feedback and recognition of the practical impact and comprehensive evaluation across datasets and settings. The reviewer raised three main concerns: robustness of cross-species testing, potential clustering confounds, and limited interpretability of attention heads. We address concerns by conducting new experiments to resolve clustering confounds.
>
>
> **Weaknesses**:
>
> - **Cross-species evaluation**: We agree that a single macaque dataset focused on a motor task is limited in task diversity, and we have clarified this in the manuscript. That said, the macaque dataset presents a meaningful domain shift, across species, regions (motor vs. hippocampal/visual), and tasks (goal-reaching vs. spontaneous/visual), making it a nontrivial test of generalization. Our results serve as proof of concept, and we plan to expand datasets and analyses in future work.
>
> - **Probe-type bias in clustering**: We thank the reviewer for raising probe-type and amplifier signatures as potential confounds. In cross-session evaluation, we held out entire sessions with different probes to avoid such biases. To assess their impact further, we ran two analyses: (1) linear probes predicting session ID, probe ID, or region; and (2) Silhouette scores for clustering by each factor. Both analyses showed that anatomical structure dominates the embeddings, with minimal influence from session or probe. See Question #2 for details.
>
> - **Interpretability and biological insight**: We agree that analyzing embedding dimensions and attention heads would provide deeper insight. We will include these visualizations in the revision.
>
> - **Preprocessing pipeline**: We thank the reviewer for pointing this out. We would like to clarify that the core preprocessing steps are described in the main text (Section 4.1, line 150-160). These steps follow the IBL LFP-band destriping pipeline, including high-pass filter, bad channel removal, rephrasing and noise reduction.
>
> - **Statistical significance**: We thank the reviewer for raising this point. We have removed Table 1 and replaced it with Supplementary Figure 6, which includes error bars to reflect variance in disease classification across folds of test sessions. We also added error bars to Figure 3 in the revised manuscript.
>
> - **Novelty and broader relevance**: While our method builds on wav2vec 2.0, it goes beyond simple domain transfer. Unlike BrainBERT (spectrogram-based) and EEG-BERT (focused on behavioral or physiological state decoding), we introduce a new task: anatomical localization from raw LFP signals at the channel level. To our knowledge, this is the first SSL approach for this problem. For a broader AI audience, our results show that general-purpose SSL models can effectively transfer to noisy, high-dimensional LFP recordings, an important step toward foundation models for neural time series.
>
> **Questions**:
>
> “Could you (a) add a within-species baseline…, and/or (b) report channel-wise confusion…?”
>
> - Thank you for the suggestion. We report channel-wise confusion in Supplementary Figure 5 and will move it to the main paper. Our macaque results are within-species: trained on macaque sessions and tested on held-out macaque sessions. This shows that our audio-pretrained model can be fine-tuned for anatomical inference in a different species. True cross-species evaluation is limited by label mismatch: mouse data focus on hippocampal/visual regions, while macaque targets motor areas. Still, we are excited to explore cross-species generalization using macaque datasets with shared labels from the Allen Institute.
>
> “Region clusters might reflect probe-type…rather than…neuroanatomy. Please test a domain-adversarial experiment … or provide a confusion table stratified by a probe model…”
>
> - We thank the reviewer for raising this important point. To minimize the risk of probe-specific signatures driving clustering, our evaluation protocol uses a strict leave-session-out strategy. This ensures that each test session comes from a different animal and probe insertion than those seen during training, reducing the chance that the model memorizes probe-related artifacts.
>
> - To directly test whether probes are encoded in the learned representations, we conducted two more analyses: (1) linear prober to predict region/session ID/probe ID from embeddings, (2) label embeddings by probe ID or session ID to assess whether they cluster by probes, session, or anatomical region
>
>
> Linear Probing Accuracy
>
> |               | Spectrogram       | SimCLR            | BrainBERT         | lfp2vec           |
> |---------------|-------------------|-------------------|-------------------|-------------------|
> | **Region**    | 0.441±0.010       | 0.683±0.053       | 0.851±0.022       | 0.921±0.004       |
> | **Session**   | 0.229±0.007       | 0.232±0.009       | 0.271±0.023       | 0.515±0.035       |
> | **Probe**     | 0.050±0.004       | 0.054±0.007       | 0.076±0.012       | 0.273±0.019       |
>
>
>
> Silhouette Score for Clustering
>
> |               | Spectrogram       | SimCLR            | BrainBERT         | lfp2vec           |
> |---------------|-------------------|-------------------|-------------------|-------------------|
> | **Region**    | -0.054±0.015      | -0.062±0.050      | 0.146±0.026       | 0.576±0.026       |
> | **Session**   | -0.077±0.022      | -0.052±0.043      | -0.024±0.006      | -0.053±0.009      |
> | **Probe**     | -0.284±0.029      | -0.121±0.093      | -0.112±0.042      | -0.266±0.006      |
>
>
> - Linear probing results show that while our model embeddings encode some session and probe identity, the dominant signal is anatomical (region) information. This suggests the representations are primarily anatomy-aware, not probe- or session-biased. Also, Lfp2vec encodes significantly more region-specific information than baseline methods.
>
> - Silhouette scores show that Lfp2vec strongly clusters by brain region, but does not cluster by probe or session, suggesting that the model learns biologically meaningful structure rather than technical artifacts. Also, Lfp2vec shows more well-separated region clusters compared to baselines.
>
> - We will add this table and visualization of embeddings colored by probe and sessions for comparison to the supplementary. We appreciate the reviewer for the excellent ideas.
>
> “Can you visualize Transformer attention heads…or perform a linear regression from embedding dimensions to canonical LFP bands?”
>
> - We agree that this is a valuable approach for interpreting model internals and have made visualization of attention heads of the last layer. We will include these visualizations in the revised manuscript.
>
>
> **Limitations**:
>
> - More detailed Alzheimer's results are shown in Supplement Figure 6. We acknowledge that the demo is preliminary in mice and not validated for clinical or diagnostic use in humans. To deploy this clinically, we would need large scale, multiple site studies, cross-species validation, comparison with clinical biomarkers, FDA approval, and human trials with reproducibility studies. We appreciate the reviewer’s concern, and will revise the broader impacts to reflect careful consideration of clinical implication of the model performance.
> - Cross-species results are shown in Supplement Figure 5. We acknowledge that this is only within species study on maceque, and performance on human subjects remains unverified and may require further changes.
> - We appreciate the reviewer for bringing up the concern regarding privacy. Though our research does not involve human recording experiments now, we recognize that privacy becomes a critical issue when scaling such approaches to human data. For future research, we could remove all direct identifiers, mask metadata, and only release pretrained models or de-identified features instead of raw LFP traces.
>
> Formatting:
> SiNAPs probe stands for Simultaneous Neural Active Pixel Sensor probe

---

> > ### Comment · Reviewer_vF5b · 2025-08-04
> >
> > I appreciate the thorough and well-executed rebuttal, especially the added experiments and clarifications. I remain supportive of the paper and will maintain my positive score.

---

### Decision · Program_Chairs · 2025-09-17

**Decision:**

Accept (poster)

**Comment:**

This paper adopts a wav2vec2.0 style modeling approach for local field potentials (LFP), evaluated on data recorded by the IBL, the Allen Visual Coding datasets, and Neuronexus datasets as well as macqque sequential reaching.

Reviewers are unamously positive about this paper. The reviewers highlighted the simplicity of transferring the wav2vec2.0 approach adapted to Lfp signals, and the the evaluation paradigm involving datasets across three labs. However, the reviewers critized the limited conceptual advance in terms of modeling. The approach is largely adopted from wav2vec 2.0, without an in-depth study if parts of the approach could be better fine-tuned to the application approach at hand. In particular, *vF5b* quotes that the broader AI relevance is moderate as the technique is a "straightforward" extension of wav2vec rather than conceptually novel (and, as my addition: the evaluation of the technique is too shallow to generate domain-specific insights beyond what the authors report in the paper). *4cyY* and *KDGD* agree regarding this. *KDGD* also criticies sparse reporting, which could partially be resolved during the rebuttal phase. Deeper analysis could have further strenghtened the manuscript.

Finally, in the results section, a lot of imprecise statements leave the discussion at a qualitative level, for instance, "sharper diagonal dominance", "Channel-wise localization accuracy" (which is not quantified in the figure), "While embeddings from SimCLR and BrainBERT show only partial clustering by region, Lfp2vec produces well-separated, compact clusters that align closely with ground-truth labels.", "especially in difficult CA2–CA3 distinctions", etc.

Considering all reviewer feedback and following the further discussion, I overall **recommend this paper for acceptance, but would ask the authors to perform the following *minor revisions* for the camera ready**:

- Work in all results promised during the rebutal phase, particular with respect to confidence intervals, and pulling results from appendix to the main paper.
- Carefully check statements in the abstract, introduction, and the results section again. In the results section, back up qualitative statements with clear summary statistics derived from the reported results

I recommend acceptance as a poster presentation. For a spotlight/oral, I agree with the reviewer's sentiment that conceptual novelty is too limited, while at the same time, a deeper empirical investigation or application to other tasks (e.g. coordinate localization, further ablations, etc.) is missing.